# A systematic review to explore how exercise-based physiotherapy via telemedicine can promote health related benefits for people with cystic fibrosis

Ben Bowhay[1], Jos M. Latour[2], Owen W. Tomlinson[3]*

1 Wye Valley NHS Trust, The County Hospital, Hereford, United Kingdom, 2 School of Nursing and Midwifery, Faculty of Health, University of Plymouth, Plymouth, United Kingdom, 3 University of Exeter Medical School, St Luke's Campus, Exeter, United Kingdom

* o.w.tomlinson@exeter.ac.uk

**Data Availability Statement:** All available data are presented in the manuscript and associated files.

**Funding:** The authors received no specific funding for this work.

## Abstract

To conduct a systematic review to evaluate the effects of physiotherapy exercises delivered via telemedicine on lung function and quality-of-life in people with Cystic Fibrosis (CF). The databases AMED, CINAHL and MEDLINE were searched from December 2001 until December 2021. Reference lists of included studies were hand-searched. The PRISMA 2020 statement was used to report the review. Studies of any design reported in the English language, included participants with CF, and within outpatient settings were included. Meta-analysis was not deemed appropriate due to the diversity of interventions and heterogeneity of the included studies. Following screening, eight studies with 180 total participants met the inclusion criteria. Sample sizes ranged from 9 to 41 participants. Research designs included five single cohort intervention studies, two randomised control trials and one feasibility study. Telemedicine-based interventions included Tai-Chi, aerobic, and resistance exercise delivered over a study period of six to twelve weeks. All included studies which measured percentage predicted forced expiratory volume in one second found no significant difference. Five studies measuring the Cystic Fibrosis Questionnaire–Revised (CFQ-R) respiratory domain found improvements, however, did not meet statistical significance. For the CFQ-R physical domain, measured by five studies, two studies found an improvement, although not statistically significant. No adverse events were reported across all studies. The included studies indicate that telemedicine-based exercise over 6–12 weeks does not significantly change lung function or quality-of-life in people with CF. Whilst the role of telemedicine in the care of pwCF is acceptable and promising; further research with standardised outcome measures, larger sample sizes and longer follow-up are required before clinical practice recommendations can be developed.

## Author summary

Preliminary evidence suggests that exercise for people with cystic fibrosis may be beneficial for lung function and quality of life improvements, but we do not know if

**Competing interests:** I have read the journal's policy and the authors of this manuscript have the following competing interests: Owen W Tomlinson and Jos M Latour report presentation fees obtained from Beam and GE Healthcare respectively.

telemedicine-based exercise can achieve similar outcomes. The current review highlighted increases within quality of life self-reported outcomes for the respiratory and physical domains, however no significant lung function improvements were found. The quality of the available evidence is low-moderate, due to poor methodological rigour and statistical power which reduces the strength of the review findings. Therefore, future large scale, high-quality randomised controlled trials with longer follow-up duration and standardised interventions would provide robust statistical evidence to optimise clinical practice and develop gold standard telemedicine-based exercise guidelines.

## 1. Introduction

Cystic Fibrosis (CF) is an autosomal recessive inherited disease which affects approximately 100,000 people worldwide and over 10,500 people in the United Kingdom (UK) [1,2]. Cystic fibrosis is a multisystem condition which predominantly impacts the respiratory system and is caused by a faulty gene code which regulates intra- and extra-cellular sodium chloride and water transference [3]. As a result, thickened dehydrated mucus builds up which leads to chronic airway infection, bronchiectasis and respiratory disease [4]. Consequently, respiratory failure is the primary cause of mortality in people with CF (pwCF) [5].

In the UK, life expectancy has increased to 47.3 years for pwCF following extensive research and treatment developments to maintain lung function [2]. Lung function is a key indicator of quality of life (QoL) and survival in pwCF [6]. The commonly used outcome measures to assess lung function and QoL in pwCF are percentage predicted forced expiratory volume in one second ($ppFEV_1$) and the Cystic Fibrosis Questionnaire–Revised (CFQ-R) [7,8]. Cystic fibrosis cannot be cured, therefore maintaining lung function and improving QoL of pwCF is a key focus [7,9,10].

The National Institute of Health and Care Excellence (NICE) guidance document [NG78] recommends that specialist physiotherapists can support pwCF by designing comprehensive, individualised exercise programmes [10]. However, no specific detail about these exercise programmes are provided in the guidelines, thus placing the onus of responsibility on to individual CF teams. Physiologically, pwCF may have reduced exercise capacity when compared against the general population due to dysfunction of their ventilatory, cardiac and muscular systems which increases their risk of mortality [11].

Interventions to increase physical activity have been developed and tested, but inconsistencies in study design and outcome measures limit our understanding of how exercise can benefit pwCF [12]. As there are no specific PA guidelines for pwCF, it is recommended that all adults should take part in at least 150 minutes of moderate vigorous intensity aerobic physical activity (MVPA) or 75 minutes of vigorous intensity physical activity per week [13].

The most common outcomes of physical activities in pwCF has been explored in various studies [14]. A multicentre study found associations between physical activity and QoL in pwCF ($n = 76$) who performed three exercise sessions per week over a 6-month study period [15]. The relationship between $ppFEV_1$ and CFQ-R respiratory domain from baseline to 6-month follow-up was found to have a significant correlation ($p < 0.01$), although $ppFEV_1$ did not have a positive correlation with the physical domain ($p < 0.05$), indicating that respiratory domains of QoL are closely linked to physical functioning [15]. A systematic review by Shelley *et al* [16]highlighted a positive relationship between $ppFEV_1$ and physical activity based on five included studies [16]. Time spent engaging in MVPA has also been positively associated with $ppFEV_1$ ($p = 0.04$) [16]. However, all these studies have been conducted face-to

face, which is a burdensome approach for CF teams. Unfortunately, there is limited evidence available of CF specific telemedicine exercise programmes which serve to maintain ppFEV$_1$ and optimise CFQ-R outcomes [17].

Telemedicine is the delivery of healthcare via technological communication methods to improve the patients' accessibility [18]. Telemedicine is a rapidly evolving provision within the National Health Service (NHS) due to an increasing number of patients living longer with chronic conditions and the current coronavirus pandemic [19]. The World Health Organization (WHO) recently published its first global strategy for digital health [20]. Telemedicine enables pwCF to access exercise programs while conforming with infection control requirements and preventing cross-infection, a requirement that restricts group exercise participation [21]. Telemedicine research for pwCF is still in the early stages, however there has been telemedicine interventions developed to support exercise in other chronic health conditions [22].

As a first step to defining optimal telemedicine-based exercise for pwCF, it is informative to review the current evidence. Currently, there are no systematic reviews exploring telemedicine-based exercise exclusively in pwCF and therefore, the aim of this systematic review was to fill this evidence gap and evaluate telemedicine-based exercise interventions for pwCF and the impact on ppFEV$_1$ and QoL.

## 2. Methods

The systematic review was conducted and reported using the PRISMA 2020 Statement [23] as a guideline.

### 2.1. Eligibility criteria

All full-text articles were obtained and screened for eligibility against the inclusion and exclusion criteria resented in **Table 1**. These eligibility criteria surrounding the population and intervention were chosen to align with the research question (i.e. only people with CF were included), and recommended physical and psychological outcome measures [10].

### 2.2. Information sources

The following databases were searched via an online research platform (EBSCOhost): AMED, CINAHL and MEDLINE; from December 2001 to December 2021. The specified date range is due to limited telemedicine interventions before this period. These databases are comprised of

**Table 1. Inclusion and exclusion criteria.**

| Selection | Inclusion criteria | Exclusion criteria |
|---|---|---|
| **Population(s)** | Cystic fibrosis patients of all ages, genders and disease severities | Non-cystic fibrosis patients |
| **Intervention (s)** | Telemedicine exercise intervention to enhance QoL and/or lung function. | Non-telemedicine interventions |
| **Comparators** | Control where possible | Not specified |
| **Outcomes** | ppFEV1 and CFQ-R physical and respiratory domains | Outcome does not relate to ppFEV$_1$ or CFQ-R domains |
| **Study design** | All study designs Published in available full-text English | Over 20 years since publication |
| **Setting** | Community-based intervention | Face-to-face interventions |

ppFEV$_1$ = Forced Expiratory Volume in 1 second; CFQ-R = Cystic Fibrosis Questionnaire–Revised; QoL = Quality of Life

healthcare research which serves to support the collation of literature relevant to the purpose of this review.

## 2.3. Search strategy

The search was developed according to three key concepts: (1) cystic fibrosis, (2) telemedicine and (3) exercise or physical activity. Keywords were identified using MeSH terms and related synonyms:

1. cystic fibrosis OR cf

2. exercise OR physical activity OR fitness OR physiotherapy

3. telemedicine OR telehealth OR telecare OR telemonitor OR telephone OR online

Boolean operators were used, with "OR" between the words in each column, and "AND" used to combine the entirety of the searches 1, 2 and 3. The reference list of included studies and review articles were also reviewed to identify any potentially relevant studies not identified through the search process.

## 2.4. Selection process

The database searches, screening of titles and abstracts was performed by one reviewer (BB) against the eligibility criteria. A second reviewer (JML) corroborated the outcome of the searches. The outcome of the included studies was confirmed by two reviewers (OT and JML).

## 2.5. Data collection process and data items

The data was extracted from each article by a reviewer (BB) into a Microsoft Excel spreadsheet and confirmed by a second reviewer (JML). The data extraction form included: authors, year of publication, country, study design, setting, participants, exercise intervention and outcome measures.

## 2.6. Study risk of bias assessment

Quality assessment of the included studies was performed by two reviewers (BB, OT) using the Critical Appraisal Skills Program (CASP) checklists for single cohort studies [24], whereby a maximal score of 12 is available; and randomised controlled studies (RCT) [25], whereby a maximal score of 11 is available. A third reviewer (JML) confirmed the results and led the discussion towards consensus of the included studies. The CASP checklist items were marked "Y" (Yes) if well described, "N" (No) if inadequate, and unclear items with a "U" (Unclear). The use of CASP as a scoring framework was chosen as this can create a score across differing designs (i.e. cohort, RCT).

## 2.7. Effect measures

Only ppFEV$_1$ and CFQ-R data was selected from the studies, as these are the most common variables obtained clinically and recommended by clinical guidelines [10]. All included studies reported ppFEV$_1$ and/or CFQ-R respiratory and physical domain outcome measures. Measures of FEV$_1$ are obtained using a spirometer and then normalised to a percentage of their predicted value. Quality of life is assessed using the CFQ-R for the respiratory and physical domains; whereby a value of 100 represented an optimal score.

## 2.8. Synthesis methods

The data of the included studies were transformed into a spreadsheet with study characteristics to provide an overview of the study methods, results and outcomes. The results of the included studies were synthesised based on the reported interventions and outcomes. Meta-analysis was not performed due to the different interventions within the included studies and no standardisation in measuring the outcomes. Therefore, the results are only synthesised based on the key findings of the outcome measures reported in the eight included studies.

# 3. Results

## 3.1. Study selection

Electronic and hand searches identified 392 studies, after removing duplicates, 346 citations were suitable for inclusion screening. Following screening of titles and abstracts, 43 full-text articles were retrieved. After assessing the articles against the inclusion criteria, eight studies were included in the review (**Fig 1**). Two of the included studies were identified through manual searching.

## 3.2. Study characteristics

Eight studies with a total of 180 pwCF were included. A summary of the included study characteristics is presented in **Table 2**.

## 3.3. Risk of bias in studies

The overall CASP quality of reporting score for the included single cohort studies ranged from 2 to 7, with a mean score of 4.6 (out of a maximum possible score of 12). The CASP scores for

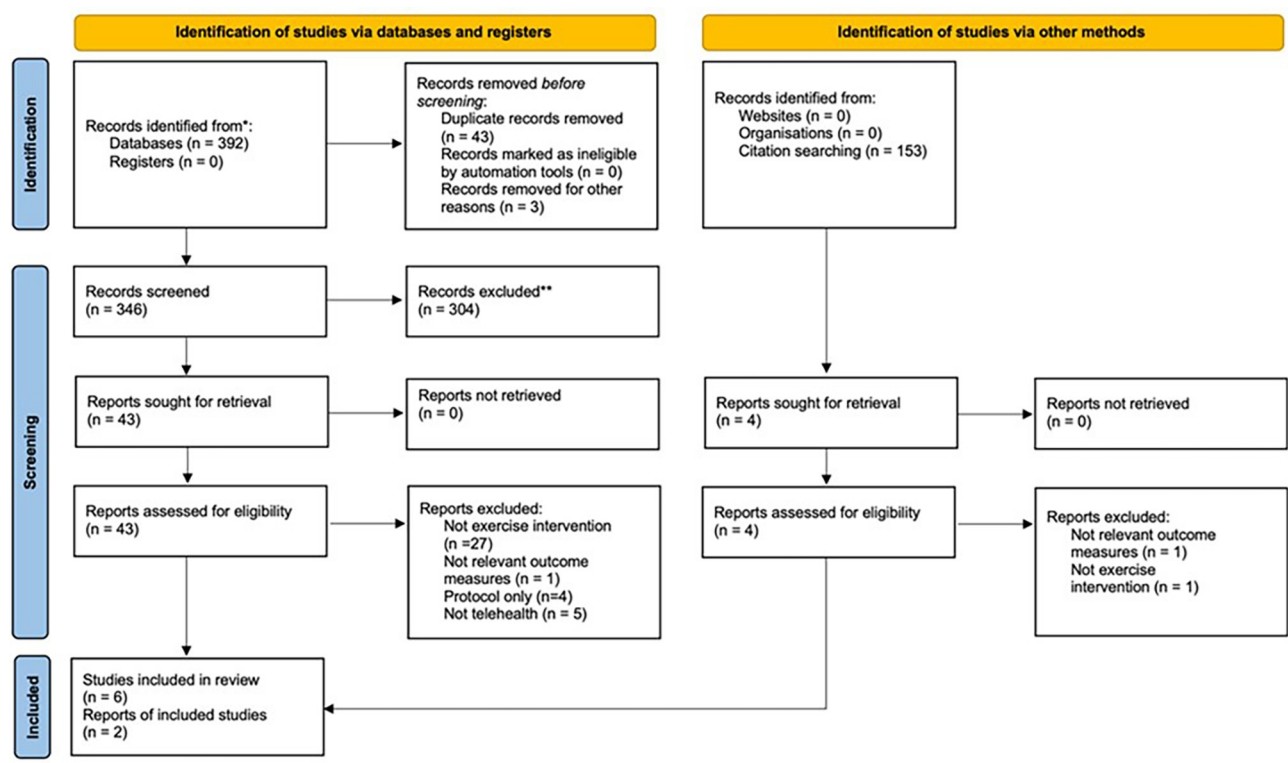

**Fig 1. PRISMA Flow Chart**

**Table 2. Study characteristics.**

| Authors and Study Design | Participant Demographics | Intervention | Comparison | Outcomes and follow up |
|---|---|---|---|---|
| **Carr et al., (2018) [32] Randomised controlled pilot study** | n = 40 Age: 6.1–51.5 years Gender = 13M, 27F CF diagnosed | Tai Chi lessons via Skype. Eight sets of Tai Chi-Qigong delivered 12w | Tai Chi lessons face-to-face; 5–10 min, max. x5 p/w. Sample analysed: I: (n = 18); C: (n = 22) | ppFEV$_1$; CFQ-R Respiratory and Physical Follow up: No follow up period Drop-outs: n = 2: intervention |
| **Chen et al., (2018) [26] Single cohort intervention study** | n = 10 Age: 8–20 years Gender = not specified CF diagnosed | Aerobic, resistance, and flexibility exercises via VSee telemedicine platform. 30-min sessions, 3x p/w, 6w | No comparators Sample analysed: n = 10 | Outcome: FEV$_1$ in litres(L) Follow up: No follow up period Drop-outs: None reported |
| **Cox et al., (2015) [29] Single cohort intervention study** | n = 10 Age: No age data available—adults Gender = 4M, 6F CF diagnosed | Internet-based physical activity via customised programme (ActivOnline). 8w total, fortnightly telephone consultation | No comparators. Sample analysed: n = 10 | Outcome: CFQ-R Respiratory and Physical Follow up: 8 weeks Drop-outs: n = 0 |
| **Hommerding et al., (2015) [27] Randomised controlled trial** | n = 34 Age: 7–20 years Gender = 20M,14F CF diagnosed | Aerobic physical exercise programme with telephone calls every 2 weeks for 12w | Control: Instructed on first day only, no follow-up. Sample analysed: n = 34 | Outcome: ppFEV$_1$ Follow up: 12 weeks Drop-outs: None reported |
| **Rovedder et al., (2014) [30] Randomised controlled trial** | n = 41 Age:(mean; SD): 24.73; 7.57 years Gender = 14M, 27F CF diagnosed | Supervised home-exercise programme via telephone for 12w | Control: Normal activities. Sample analysed: n = 41 | Outcome: ppFEV$_1$ Follow up: 12 weeks–unclear? Drop-outs: None reported |
| **Schmidt et al., (2011) [28] Single cohort intervention study** | n = 24 Age: 14–40 years Gender = 15M, 9F CF diagnosed | Individualised exercise programme with weekly phone call. +30min, x3 p/w, 12w. | Control. Sample analysed: I: (n = 14); C: (n = 10) control (dropouts) | Outcome: ppFEV$_1$ Follow up: No follow up period. Drop-outs: n = 10. |
| **Swisher et al., (2010) [31] Single cohort intervention study** | n = 12 Age: 7–17 years Gender = 5M, 7F CF diagnosed | Activity-based intervention with weekly follow-up phone calls for 12w. 10,000 step/day target | No comparators. Sample analysed: n = 12 | Outcome: CFQ-R Respiratory and Physical Follow up: No follow up period Drop-outs: n = 2 |
| **Tomlinson et al., (2020) [33] Single cohort intervention study** | n = 9 Age: 15.5–42.1 years Gender = 6M, 3F CF diagnosed | Individualised exercise intervention conducted via Skype for 8w, supervised by an exercise therapist. | No comparators. Sample analysed: n = 7 | Outcome: ppFEV$_1$, CFQ-R Respiratory and Physical Follow up: 4 weeks Drop-outs: n = 2 |

I = intervention; C = control; M = male; F = female; w = week; p/w = per week. CFQ-R = Cystic Fibrosis Questionnaire–Revised; ppFEV$_1$ = Forced Expiratory Volume in 1 second.

the RCTs ranged from 6 to 7, with a mean score of 6.5 (out of a maximum possible of 11), as shown in **Tables 3 and 4**. The most common reasons for a low CASP score in the cohort studies were inaccurate measure of exposure and outcome to minimise bias, poor identification of confounding factors and inadequate follow up. Whereas, within the RCTs low scores were found due to inadequate blinding of participants, and poor reporting of the use of confidence intervals. The majority of studies included all post intervention outcome measures, however, one study did not report individual patient data for the assessed outcomes due to the risk of participant identification [26]. Full outcome datasets were not reported in three of the eight studies [26–28].

## 3.4. Synthesis of result

**3.4.1. Study characteristics.** The eight included studies consisted of 180 pwCF in total, of which 77 participants were male, however, one study had no gender ratio data available [26].

**Table 3. CASP Scores for Single Cohort Studies.**

| Study | Item 1 | Item 2 | Item 3 | Item 4 | Item 5 | Item 6 | Item 7 | Item 8 | Item 9 | Item 10 | Item 11 | Item 12 | Total |
|---|---|---|---|---|---|---|---|---|---|---|---|---|---|
| **Chen et al., 2018 [26]** | Y | U | U | N | N | N | N | N | Y | Y | Y | U | 4 |
| **Cox et al., 2015 [29]** | Y | Y | U | N | N | N | N | N | Y | Y | Y | U | 5 |
| **Schmidt et al., 2011 [28]** | Y | U | N | N | N | N | N | N | U | U | Y | U | 2 |
| **Swisher et al., 2010 [31]** | Y | Y | N | N | N | N | N | N | Y | Y | Y | U | 5 |
| **Tomlinson et al., 2020 [33]** | Y | Y | N | N | N | Y | N | N | Y | Y | Y | Y | 7 |

CASP: Critical Appraisal Skills Program. Y: Yes; N: No; U: Unclear.

Two studies recruited adults only [29,30], three studies aimed to recruit children and adolescents [26,27,31] and two studies included participants from 6.1–51.5 years [28,32]. Sample sizes ranged from 9 to 41. Study designs included five single cohort intervention studies [26,28,29,31,33], two RCTs [27,30] and one feasibility trial [32]. Five studies did not have a control group [26,28,29,32,33].

**3.4.2. Intervention characteristics.** The method of telemedicine delivery was variable across studies (**Table 2**) with three online video-calling [26,32,33] and five telephone-based trials [27–31]. Intervention duration ranged from 6–12 weeks, with many of the included studies lasting 12 weeks [27,28,30,31,33]. Exercise programmes were supervised in real-time for 3/8 studies [26,32,33], and unsupervised with telephone follow ups in 5/8 studies [27–31].

The frequency of exercise varied from once a week to daily. Intensity of exercise was only stated in three studies [26,28,33]. Exercise session duration and intensity did not meet physical activity guidelines for 4/8 of the included studies [26,29,32,33] and was not reported in 3/8 studies [27,30,31]. The type of exercise differed within the included studies; seven studies used a combination of aerobic exercises, rather than focussing on one method [26–31,33] (**Table 2**). Follow-up was often not performed [26,28–31], participant withdrawals ($n = 2$–10) occurred in four studies [32,28,31,33] and two studies did not report withdrawals [26,27].

**3.4.3. Outcomes.** The main findings of the outcome measures are presented in **Table 5**. All included studies which measured $ppFEV_1$ found no significant difference [26–28,30,32,33]. However, the majority revealed a slight reduction in $ppFEV_1$ [26–28,30,33]. The validated CFQ-R was reported in five studies to provide an insight into patients perceived health [28–31,33]. Five studies found improvements in the CFQ-R respiratory domain. For the CFQ-R physical domain, two studies found an improvement [29,30], and three studies showed reduced scores [28,31,33].

## 4. Discussion

The aim of this review was to investigate the effectiveness of telemedicine exercise interventions on $ppFEV_1$ and QoL in pwCF. The eight included studies demonstrated that a variety of

**Table 4. CASP Scores for Randomised Control Trials.**

| Study | Item 1 | Item 2 | Item 3 | Item 4 | Item 5 | Item 6 | Item 7 | Item 8 | Item 9 | Item 10 | Item 11 | Total |
|---|---|---|---|---|---|---|---|---|---|---|---|---|
| **Carr et al., 2018 [32]** | Y | Y | Y | N | Y | Y | N | N | U | Y | Y | 7 |
| **Hommerding et al., 2015 [27]** | Y | Y | Y | N | Y | Y | N | N | U | Y | Y | 7 |
| **Rovedder et al., 2014 [30]** | Y | Y | Y | N | Y | Y | N | N | U | Y | N | 6 |

CASP: Critical Appraisal Skills Program. Y: Yes; N: No; U: Unclear.

**Table 5. Key findings of the outcome measures in the eight included studies.**

| Authors | Control | ppFEV$_1$ | CFQ-R Physical domain | CFQ-R Respiratory domain |
|---|---|---|---|---|
| **Carr et al., (2018) [32]** | Standard care control group | Increased 1.47% predicted | No significant differences reported | No significant differences reported |
| **Chen et al., (2018) [26]** | No comparators | I:pre 85L vs I:post 81.5L*** | Not reported | Not reported |
| **Cox et al., (2015) [29]** | No comparators | Not reported | I:pre 36 (32–67) vs I:post 63 (29–79) ($p = 0.2$) | I:pre 33 (27–70) vs I:post 44 (33–71) ($p = 0.2$) |
| **Hommerding et al., (2015) [27]** | Standard care control group | I:pre (95.5+/-17.9) vs I:post (93.7 +/-8.6) C:pre (100.1+/-21.2) vs C:post (99.1 +/-14.2) ($p = 0.5$) | Not reported | Not reported |
| **Rovedder et al., (2014) [30]** | Standard care control group | I:pre (58.3+/-27.6) vs C:pre (57.6 +/-22.7) ($p = 0.709$) I:post (52.3 +/-16.1) vs C:post (55.6 +/-7.3) ($p = 0.306$) | I:pre 58 (*45/87) vs C:pre 64 (*44/81) ($p = 1.000$) I:post (64.1+/-17.50) vs C:post 2.4+/-17.50 (*-10/13) ($p = 0.742$) | I:pre 55 (*50/72) vs C:pre 55 (*38/61) ($p = 0.311$) I:post **3.8+/-10.60 (*0/11) vs C:post **-4.7+/-13.40 (*-1/7) ($p = 0.925$) |
| **Schmidt et al., (2011) [28]** | No comparators | I:pre 87.6 (74.8;105.8) I:post 85.0 (76.5;99.0) ($p = 0.377$) | I:pre (90.8+/-14.5) vs I:post (93.8+/-8.1) ($p = 0.346$) | I:pre (77.8+/-12.7) vs I:post (76.2+/-13.5) ($p = 0.371$) |
| **Swisher et al., (2010) [31]** | No comparators | Not reported | I:pre (86.1+/-10.4) vs I:post (81.8+/-15.0) | I:pre (77.5+/-10.2) vs I:post (84.7+/-10.1) |
| **Tomlinson et al., (2020) [33]** | No comparators | I:pre (74+/-31) vs I:post (73+/-34) | I:pre 72 (34) vs I:post 58 (37). | I:pre 52 (33) vs I:post 56 (23) |

I = intervention; C = control; pre = pre-intervention; post = post-intervention; *Author did not report full outcome dataset PRE vs POST; +/- = standard deviation; * = Median 25/75%; ** = difference;***ppFEV$_1$ not reported–we query this value as Litres.

telemedicine-based exercises might have some benefit for pwCF. Improvements were found in the CFQ-R respiratory domain in five studies indicating a beneficial impact on perceived QoL [28–31,33]. However, no significant benefits were reported on ppFEV$_1$ which is a gold standard measure of respiratory system performance.

## 4.1. Interventions

There was substantial heterogeneity in the design of studies, and how these have each implemented interventions. For example, all studies using video-calling were published more recently from 2018–2020 [26,32,33], whereas studies testing telephone interventions were completed many years prior. This clearly shows the evolution towards online-based interventions within CF healthcare and a shift away from traditional methods. The COVID-19 pandemic has further accelerated the acceptance and use of telemedicine in pwCF and their clinical teams [34], as well as the general population. This aligns with the long term plans of the UK National Health Service, whereby the role of technology and telehealth is anticipated to substantially grow in the coming years [35,36].

The frequency of telemedicine guided exercises varied from once a week to daily and the intensity of exercise was only stated in three studies [26,28,33]. The majority of the studies utilised a combination of aerobic exercises, rather than focussing on one method [26–31,33], and this heterogeneity in programmes may be due to the unique characteristics of pwCF, consolidating how they may require more specific individualised exercise programmes; whereby individualisation of programmes has been recently advocated for by international experts in exercise management of CF [37].

Within this review, the shorter interventions (6–8 weeks) were online-based [26,29,33], and thus within these studies, decreased ppFEV1 outcomes were reported, a finding corroborated by Radtke *et al* [38], who indicated that short-term conventional exercise interventions for pwCF did not show differences in outcomes. Moreover, only one study in our review met the WHO physical activity recommendations with over 75 minutes of vigorous intensity physical activity performed weekly. This suggests that the majority of interventions may not have been adequate to promote pulmonary function, and therefore, this intensity (or lack thereof), alongside the impact of intervention length, and the aforementioned variation in study design, means we cannot determine the efficacy of specific exercises at present. Therefore, longer study durations with rigorous follow-up may be required for telemedicine-based exercise interventions with people with CF.

## 4.2. Changes in function

The $ppFEV_1$ findings within this review do not align with prior research in pwCF, which highlights $ppFEV_1$ improvements following conventional exercise interventions [15]. On the other hand, a $ppFEV_1$ decline of approximately -1.52% per year (95% CI: -1.66 to -1.38%) has been highlighted within the UK CF registry [39]. The minimal decline found within this review aligns with anticipated CF disease progression, therefore a focus on maintaining $ppFEV_1$ may be more appropriate. The observed decreased of $ppFEV_1$ results in our review may also be because the tested interventions might not meet the physiological requirements for pwCF specifically.

The CFQ-R respiratory domain outcomes in our review highlights that pwCF may perceive exercise as beneficial for their respiratory system function, despite the lack of improvement in objective $ppFEV_1$ performance; thus highlighting the importance of including both objective, physical function, and subjective, psychological function, when evaluating interventions in disease groups. Interestingly, for the CFQ-R respiratory domain, reduced adherence was reported in both of the included studies that showed reduced scores [31,33], and within the wider CF population, it has been reported that adherence to conventional exercise programmes has been suboptimal [9,40,41]. For example, Rovedder *et al* [30] indicated that higher QoL scores were related to participants who performed more frequent telemedicine-based exercise, and highlighted the importance of adherence to tailored long-term programmes that meets the specific needs and preferences of pwCF.

The CFQ-R results of the included studies in our review testing telemedicine-based exercise are not consistent with conventional face-to-face exercises as reported by Hebestreit *et al* [15] who showed no change in the CFQ-R physical and respiratory domains among pwCF. Statistical significance was observed in our review and also not in the study of Hebestreit *et al* [15] This may be due to small sample sizes, different study aims and insufficient statistical power which limits the analysis of data to descriptive means.

## 4.3. Strengths & limitations

Whilst many strengths exist, notably this being the first study to systematically pool data on exercise-based telehealth interventions in this unique population, limitations of this review must be acknowledged. In the design of this review, grey literature was not assessed, non-English language articles were not included, and not all available databases were comprehensively searched, with only EBSCOhost used. Moreover, only a single-screener was used to filter results, which may introduce bias, although proportion of 'missed' studies in single-screened reviews is low (median of 5%) [42]. Another limitation of this review is the small sample sizes and heterogeneity in study designs and outcomes, thus precluding meta analyses and therefore

limiting the strength of the conclusion and recommendations. Only eight studies were included within the review, despite a comprehensive search strategy, this highlights that there is currently limited evidence available. Generalisability and transferability of the results to clinical practice is limited due to methodological differences and variation in the study population and interventions.

## 5. Conclusion

The current review highlighted promising CFQ-R improvements within the respiratory and physical domains following 6–12 weeks of telemedicine-based exercise. However, no significant changes were found in ppFEV$_1$ outcomes. This suggests that pwCF may find telemedicine-based exercise beneficial for enhancing their QoL, despite having no clear pulmonary function improvements.

The implication of this review on clinical practice and research is that caution should be applied when evaluating the evidence within the emerging field of telemedicine-based exercise for pwCF due to the methodological shortcomings and poor data quality within the included studies. Further large scale, high-quality RCTs with longer follow-up periods and standardised interventions would elucidate the impact of telemedicine-based exercise on ppFEV1 and QoL outcomes for pwCF further. The challenges will be recruitment to increase sample sizes, and provide robust statistical evidence that informs evidence-based guidelines and clinical practice. However, as this modality of intervention delivery is likely now a permanent fixture for CF in a post-pandemic landscape [34], it is likely that higher quality interventions will be developed and implemented in years to come.

## Supporting information

**S1 PRISMA Checklist. PRISMA Checklist**
(PDF)

## Acknowledgments

The authors thank Andrew Sawyer, Research Advisor at Birmingham City University in Birmingham, UK, for his contribution to the search strategy development.

## Author Contributions

**Conceptualization:** Ben Bowhay.

**Data curation:** Ben Bowhay, Jos M. Latour.

**Formal analysis:** Ben Bowhay, Jos M. Latour, Owen W. Tomlinson.

**Methodology:** Ben Bowhay, Jos M. Latour, Owen W. Tomlinson.

**Project administration:** Ben Bowhay.

**Supervision:** Jos M. Latour, Owen W. Tomlinson.

**Writing – original draft:** Ben Bowhay.

**Writing – review & editing:** Ben Bowhay, Jos M. Latour, Owen W. Tomlinson.

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
