## [Decision Letter · Decision Letter 0]

8 Nov 2022

PDIG-D-22-00220

A systematic review to explore how exercise-based physiotherapy via telemedicine can promote health related benefits for people with cystic fibrosis

PLOS Digital Health

Dear Dr. Tomlinson,

Thank you for submitting your manuscript to PLOS Digital Health. After careful consideration, we feel that it has merit but does not fully meet PLOS Digital Health's publication criteria as it currently stands. Therefore, we invite you to submit a revised version of the manuscript that addresses the points raised during the review process.

Please submit your revised manuscript within 30 days Dec 08 2022 11:59PM. If you will need more time than this to complete your revisions, please reply to this message or contact the journal office at digitalhealth@plos.org. Please include the following items when submitting your revised manuscript:

We look forward to receiving your revised manuscript.

Kind regards,

Yuan Lai, Ph.D.

Academic Editor

PLOS Digital Health

Journal Requirements:

1. We have amended your Competing Interest statement to comply with journal style. We kindly ask that you double check the statement and let us know if anything is incorrect. 

2. Please send a completed 'Competing Interests' statement, including any COIs declared by your co-authors. If you have no competing interests to declare, please state "The authors have declared that no competing interests exist". Otherwise please declare all competing interests beginning with the statement "I have read the journal's policy and the authors of this manuscript have the following competing interests:"

Additional Editor Comments (if provided):

Reviewers' comments:

Reviewer's Responses to Questions

**Comments to the Author**

1. Does this manuscript meet PLOS Digital Health’s publication criteria? Is the manuscript technically sound, and do the data support the conclusions? The manuscript must describe methodologically and ethically rigorous research with conclusions that are appropriately drawn based on the data presented.

Reviewer #1: Partly

Reviewer #2: Yes

Reviewer #3: Yes

2. Has the statistical analysis been performed appropriately and rigorously?

Reviewer #1: N/A

Reviewer #2: N/A

Reviewer #3: Yes

3. Have the authors made all data underlying the findings in their manuscript fully available (please refer to the Data Availability Statement at the start of the manuscript PDF file)?

Reviewer #1: Yes

Reviewer #2: Yes

Reviewer #3: Yes

4. Is the manuscript presented in an intelligible fashion and written in standard English?

Reviewer #1: Yes

Reviewer #2: Yes

Reviewer #3: Yes

5. Review Comments to the Author

Reviewer #1: Below I provide section by section, critique of the shortcomings of the manuscript: 

Eligibility Criteria: Some justification is needed for how and why the eligibility criteria was designed to be how it is. 

Justification is needed for why only three databases searched and why were these three databases selected. specifically, PUBMED seems to be concerningly absent from the list of databases

Selection Process: "A second reviewer (JML) performed the outcome of the searches" -- meaning of this sentence is unclear.

Selection Process: Authors need to address the fact that the initial search and screening was only conducted by one person -- this is highly unusual for systematic reviews. this needs to be addressed as a limitation of this study as this is major source of bias in this review.

Study Risk of Bias Assessment: A risk bias assessment of the review itself needs to be conducted and reported. Consider ROBIS: https://www.ncbi.nlm.nih.gov/pmc/articles/PMC4687950/

Effect Measures: Why were only ppFEV and CFQ-R data selected from the study? Simply the fact that all studies reported these is a shallow justification and does not give the reader a robust understanding of this decision.

3.3. Risk of Bias in Studies: The authors should provide more context for what the CASP score means; a mean score of 4.6 for single cohort studies does not mean much to a reader not familiar with CASP. 

3.4. Synthesis of Results: Authors should consider using sub-sections to structure their results synthesis section and thus adding more hierarchy to their reported results and framing the findings of the review more clearly. 

Authors should recognize that results of a review are not simply the outcomes of the reviewed studies but also their intervention and experimental designs. For example, In the results section, the authors should also report on how user adherence to telemedicine programs were measured and how this impacted the outcomes of the reviewed studies. 

4. Discussion: The authors should use sections to structure their discussion and frame their argument. 

The discussion sections requires significant work to be publishable. Authors need to clearly report on the state of this topic, address specific shortcomings, comment on the impact of the shortcomings on the lackluster results of the reviewed studies and suggest robust guidelines for future work in this area. Currently, the discussion sections reads more like a results sections, than their current results section. For instance, there needs to be discussion on the contents of the various Tables provided in this section in context of their impact on the reviewed studies and how they can be used to inform improved future studies.

Reviewer #2: Thank you for this interesting and important topic, your manuscript is organized and written very well, and the discussion is strong. The only concern is that your manuscript sounds like a literature review of 8 articles but as you mentioned that it was because of a lack of enough studies in the literature.

Reviewer #3: The health-related benefits of exercise-based physiotherapy via telemedicine is an important question to the CF community at this time. Post-pandemic, in CF centres across the UK physiotherapists continue to spend time developing and carrying out online-exercise programmes for people with CF in order to meet the changing needs of the CF population. It is so important that we begin to discuss and assess the value of these programmes so that we can begin to define what a quality CF physiotherapy telemedicine exercise programme looks like to inform our future strategy. This paper will therefore have broad interest within the CF community.

The report shows excellent methodological design which is described in enough detail to be reproduced in future reviews.

The conclusions are appropriate for the report's findings.

The report is well organised, easy to read and understand.

6. PLOS authors have the option to publish the peer review history of their article (what does this mean?). If published, this will include your full peer review and any attached files.

**Do you want your identity to be public for this peer review?** For information about this choice, including consent withdrawal, please see our Privacy Policy.

Reviewer #1: No

Reviewer #2: Yes: Nisreen Al Jallad

Reviewer #3: Yes: Catherine Brown

---

## [Decision Letter · Decision Letter 1]

24 Jan 2023

A systematic review to explore how exercise-based physiotherapy via telemedicine can promote health related benefits for people with cystic fibrosis

PDIG-D-22-00220R1

Dear Mr Tomlinson,

We are pleased to inform you that your manuscript 'A systematic review to explore how exercise-based physiotherapy via telemedicine can promote health related benefits for people with cystic fibrosis' has been provisionally accepted for publication in PLOS Digital Health.

Best regards,

Yuan Lai, Ph.D.

Academic Editor

PLOS Digital Health

Reviewer Comments (if any, and for reference):

Reviewer's Responses to Questions

**Comments to the Author**

1. If the authors have adequately addressed your comments raised in a previous round of review and you feel that this manuscript is now acceptable for publication, you may indicate that here to bypass the “Comments to the Author” section, enter your conflict of interest statement in the “Confidential to Editor” section, and submit your "Accept" recommendation.

Reviewer #1: All comments have been addressed

Reviewer #2: All comments have been addressed

2. Does this manuscript meet PLOS Digital Health’s publication criteria? Is the manuscript technically sound, and do the data support the conclusions? The manuscript must describe methodologically and ethically rigorous research with conclusions that are appropriately drawn based on the data presented.

Reviewer #1: Yes

Reviewer #2: Yes

3. Has the statistical analysis been performed appropriately and rigorously?

Reviewer #1: N/A

Reviewer #2: Yes

4. Have the authors made all data underlying the findings in their manuscript fully available (please refer to the Data Availability Statement at the start of the manuscript PDF file)?

Reviewer #1: Yes

Reviewer #2: Yes

5. Is the manuscript presented in an intelligible fashion and written in standard English?

Reviewer #1: Yes

Reviewer #2: Yes

6. Review Comments to the Author

Reviewer #1: Thank you for addressing my previous comments. I believe the current versions of the manuscript represents a significant improvement over the previous version. Thank you for all your hardwork in making these improvement; It will be a great resource for furthering future research in this domain.

Reviewer #2: Well done

7. PLOS authors have the option to publish the peer review history of their article (what does this mean?). If published, this will include your full peer review and any attached files.

**Do you want your identity to be public for this peer review?** For information about this choice, including consent withdrawal, please see our Privacy Policy.

Reviewer #1: No

Reviewer #2: **Yes: **Nisreen Al Jallad
